

# Soymilk residue (okara) as a natural immobilization carrier for *Lactobacillus plantarum* cells enhances soymilk fermentation, glucosidic isoflavone bioconversion, and cell survival under simulated gastric and intestinal conditions

Xia Xiudong, Wang Ying, Liu Xiaoli, Li Ying and Zhou Jianzhong

Institute of Agro-Product Processing, Jiangsu Academy of Agricultural Sciences, Nanjing, Jiangsu, PR China

Corresponding author
Zhou Jianzhong,
1939650782@qq.com

## ABSTRACT

Cell immobilization is an alternative to microencapsulation for the maintenance of cells in a liquid medium. However, artificial immobilization carriers are expensive and pose a high safety risk. Okara, a food-grade byproduct from soymilk production, is rich in prebiotics. Lactobacilli could provide health enhancing effects to the host. This study aimed to evaluate the potential of okara as a natural immobilizer for *L. plantarum* 70810 cells. The study also aimed to evaluate the effects of okara-immobilized *L. plantarum* 70810 cells (IL) on soymilk fermentation, glucosidic isoflavone bioconversion, and cell resistance to simulated gastric and intestinal stresses. Scanning electron microscopy (SEM) was used to show cells adherence to the surface of okara. Lactic acid, acetic acid and isoflavone analyses in unfermented and fermented soymilk were performed by HPLC with UV detection. Viability and growth kinetics of immobilized and free *L. plantarum* 70810 cells (FL) were followed during soymilk fermentation. Moreover, changes in pH, titrable acidity and viscosity were measured by conventional methods. For *in vitro* testing of simulated gastrointestinal resistance, fermented soymilk was inoculated with FL or IL and an aliquot incubated into acidic MRS broth which was conveniently prepared to simulate gastric, pancreatic juices and bile salts. Survival to simulated gastric and intestinal stresses was evaluated by plate count of colony forming units on MRS agar. SEM revealed that the lactobacilli cells attached and bound to the surface of okara. Compared with FL, IL exhibited a significantly higher specific growth rate, shorter lag phase of growth, higher productions of lactic and acetic acids, a faster decrease in pH and increase in titrable acidity, and a higher soymilk viscosity. Similarly, IL in soymilk showed higher productions of daizein and genistein compared with the control. Compared with FL, IL showed reinforced resistance to simulated gastric and intestinal stresses *in vitro* that included low pH, low pH plus pepsin, pancreatin, and bile salt. Our results indicate that okara is a new potential immobilization carrier to enhance the growth and glucosidic isoflavone bioconversion activities of *L. plantarum* in soymilk and improve cell survivability following simulated gastric and intestinal conditions.

## INTRODUCTION

Developing novel foods containing probiotics has attracted increasing interest in recent years. The Food and Agriculture Organization and World Health Organization defined probiotics as "live microorganisms which when administered in adequate amounts confer a health benefit to the host." *Lactobacillus* and *Bifidobacterium*, originally isolated from the human intestine, are the most widely used probiotics. Probiotics provide many health benefits, such as prevention of pathogenic infections, maintenance of intestinal microbial homeostasis, alleviation of lactose intolerance, enhancement of immune response, stabilization of gastrointestinal (GI) barrier function, and production of anti-mutagenic and anti-carcinogenic compounds (*Choi et al., 2006*; *Boirivant & Strober, 2007*; *Panthapulakkal & Sain, 2007*; *Saulnier et al., 2009*).

Probiotics must contain a sufficient amount of live bacteria (at least $10^6$–$10^7$ CFU/g) to deliver health benefits (*Boylston et al., 2004*). Probiotics do not always survive under the acidic conditions of the upper GI tract to proliferate in the intestine. Several methods have been proposed to improve the viability of probiotics, and cell immobilization appears to be the most promising among these methods (*Cai et al., 2014*; *Sathyabama et al., 2014*). Cell immobilization, which refers to the entrapment of biomass within various supports, has been widely used to increase the growth, stability, and viability of microorganisms (*Teh et al., 2010*). This technology has been largely applied in the pharmaceutical (e.g., drug and vaccine delivery) and agricultural sectors (e.g., fertilizers). In addition, cell immobilization has been poised to provide immense benefits to the food industry (*Champagne, Lee & Saucier, 2010*).

Compared with fermentation with free cells, fermentation with immobilized cells show higher fermentation rates, better substrate utilization, lower cost, less product inhibition, more favorable microenvironment to the cell, and other benefits (*Indira et al., 2015*). The performance of immobilized cell system depends on the right selection of the immobilization supports (*Genisheva et al., 2011*). Gel entrapment techniques have been widely used in cell immobilization on laboratory and industrial scales, while one disadvantage of gel matrices is that they hinder substrate diffusion to and metabolite release from immobilized cells (*Guénette, & Duvnjak, 1996*).

Cell immobilization is beneficial for the food industry (*Kourkoutas et al., 2005*). Many efforts have focused on the immobilization of probiotics within various natural supports, such as fruit pieces (*Kourkoutas et al., 2005*; *Kourkoutas et al., 2006*), starch (*Mattila-Sandholm et al., 2002*), casein (*Dimitrellou et al., 2009*), wheat grains (*Bosnea et al., 2009*), agro-wastes (*Teh et al., 2010*), *Pistacia terebinthus* resin (*Schoina et al., 2015*), and bacterial cellulose (*Fijałkowski & Peitler, 2015*). These studies have aimed to stabilize cells and formulate new types of foods fortified with immobilized probiotics released more in the human gut.

Soymilk residues, also known as okara, are the by-products of soymilk and tofu processing. Okara is rich in dietary fibres (mainly insoluble dietary fibre), proteins, sugars, lipids, isoflavones, minerals and vitamins (*O'Toole, 1999*; *Villanueva-Suárez et al., 2013*). Different processing methods can have an influence in okara composition and profound effects on the microbiological quality. In the Chinese processing method, soybeans are soaked, rinsed, and ground, and the okara is filtered off with no heat treatment; In contrast, the Japanese processing method uses heat treatment to the soaked and ground mixture, and the okara contained less amounts of soluble fibre components and more protein, because of a possible coagulation induced by heating, but have no influence in the insoluble dietary fibres (*Espinosamartos & Rupérez, 2009*). Because of the nutritive valve, blank taste and the absence of color, okara is suitable for the production of functional foods (*Villanueva-Suárez et al., 2013*). It has been confirmed that okara as a diet supplement shows the function of preventing obesity, increasing antioxidant status, reducing plasma cholesterol, maintaining a balanced intestinal flora, anti-hypertension and increasing calcium absorption and retention, etc (*O'Toole, 1999*; *Jiménez-Escrig et al., 2008*; *Pérezlópez et al., 2016*). *Villanueva et al. (2011)* found that dietary fibre and protein of okara could be related with the total lipids and cholesterol decrease in the plasma and liver, as well as with the faecal output increase in high-fat fed hamsters. The colon is a complex system that includes not only human tissue but also a complex microbiota population. *Lactobacillus acidophilus* and *Bifidobacterium bifidum* are both identified as probiotics of the human intestinal microbiota. The research of *Préstamo et al. (2007)* showed that in okara-fed rats, *in vivo* colonic fermentation of okara resulted in a lower pH, but a higher cecal weight and higher total short chain fatty acids production, and short chain fatty acids are responsible for the low pH in the cecum. *Pérezlópez et al. (2016)*, whose research indicated that okara exhibited potential prebiotic effect, which inoculated with human faecal slurries could increase the growth of bifidobacteria and lactobacilli, and inhibit the growth of potentially harmful bacterial. Moreover, attributed to the great complexity of okara's cell wall, which would need longer times to be fermented than other easily digested molecules, thus allowing an extended potential prebiotic effect.

According to *Grizotto & Aguirre (2011)*, approximately 2–3 tons of okara are produced per ton of processed soybean. As a result, more than 2,800,000 tons of soymilk residues are generated annually in China (*Zhu et al., 2012*). Only a small amount is used to produce feed and fertilizer while the rest are discarded, leading to serious environmental issues. Therefore, technologies that utilize okara are urgently needed. At present, no study has attempted to utilize okara as an immobilization support for lactic acid bacteria (LAB). The survival and viability of LAB immobilized on okara under simulated gastrointestinal condition also remain unknown. The present study aimed to evaluate okara's potential as a *L. plantarum* immobilizer and to examine the growth and metabolic characteristics of okara-immobilized *L. plantarum* in soymilk. We also assessed the survival of okara-immobilized *L. plantarum* cells under simulated gastric and intestinal conditions.

## MATERIALS AND METHODS

### Bacterial culture

*L. plantarum* 70810 was obtained from the Laboratory of Food Microbiology, College of Food Science and Technology, Nanjing Agricultural University. The stock culture was stored at −20 °C in 40% (v/v) sterile glycerol. This strain was propagated three times in sterile de Mann, Rogosa, Sharpe (MRS) broth (Aobox, Beijing, China) and incubated at 37 °C for 20 h prior to use.

### Preparation of soymilk and okara

Dried soybeans purchased from Suguo market (Nanjing, Jiangsu, China) were rinsed and soaked in distilled water for approximately 12 h at room temperature. The macerated beans were drained and ground with distilled water (water:dry bean ratio of 9:1) in a grinder (JYL-C022E, Joyoung, China). The blended mixture was filtered with a muslin cloth to collect soymilk and okara. Soymilk was pasteurized at 95 °C for 15 min for producing fermented soymilk. The oakra was washed three times and dried in an oven (TY-HX-SY-04; Suzhou City Taiyu Oven Equipment CO., LTD, China) at 70° to a constant weight. The okara was further milled with a mill (JP-300A-8; Yong kang Jiu pin Industry and Trade Co., Ltd., China) and sieved through a 120 test sieve. The resultant powder was vacuum-packed and stored at −20 °C until further use.

### Preparation of free (FL) and okara-immobilized (IL) cells

*L. plantarum* 70810 was cultivated statically in 100 mL of MRS broth (Aobox, Beijing, China) at 37 °C for 24 h. Cells were centrifuged at 12,000× g for 15 min at 4 °C. Pellets were washed three times with sterile saline solution (0.85% NaCl, w/v). Immobilization was performed as follows: 4% (w/v) okara powder was added to triangular flasks containing 100 mL of MRS broth and autoclaved at 121 °C for 15 min. Afterward, 3% (v/v) activated cultures were transferred aseptically into the MRS broth containing okara powder and fermented at 37 °C for 24 h. When immobilization was completed, the fermented medium was filtered through cheese cloth to harvest the immobilization supports retained on the cloth. IL were washed three times and used as a starter for soymilk fermentation or for *in vitro* GI stress tolerance tests.

### Scanning electron microscopy (SEM)

MRS broth containing IL was centrifuged at 2,300× g for 5 min. The obtained pellets were washed five times with sterile saline solution (0.85% NaCl, w/v). The pellets were resuspended in 3.5% glutaraldehyde for 6 h; dried by treatment with 50%, 70%, 90%, 95%, and 100% ethanol; and then stored overnight in a desiccator to remove moisture. The samples were coated with gold and examined under a scanning electron microscope (EVO-LS10; Carl Zeiss, Oberkochen, Germany). Sterilized okara without *L. plantarum* 70810 was used for comparison.

### Inoculation of FL and IL into soymilk

The prepared FL and IL cells were centrifuged at 12,000× g for 15 min at 4 °C. Pellets containing *L. plantarum* 70810 cells were inoculated aseptically in 100 mL of soymilk.

Fermentation processes were performed at 37 °C for 8 h. The fermented soymilk was used for chemical analysis and *in vitro* GI stress tolerance tests.

## Ultrasonic treatment and viable cell count

Viability test of *L. plantarum* 70810 strains was conducted as previously reported with slight modifications (*Teh et al., 2010*). Cell shedding from okara was performed with an ultrasonic cleaner (YQ-520C; Shanghai Yijing Ultrasonic Instrument Co., Ltd., China) under an ultrasound power of 160 W for 10 min at initial temperature of 20 °C. The samples were diluted ($10^{-1}$–$10^{-6}$) with sterile saline solution (0.85% NaCl, w/v), and a 100 μL sample was dropped onto MRS agar plates. Individual colonies were counted after 48 h of incubation at 37 °C. Viable cell counts were calculated as log colony-forming units per gram (log CFU/g).

## Analysis of microbial growth kinetics of *L. plantarum* 70810 in soymilk

The microbial growth kinetics of FL or IL in soymilk was calculated using a modified Gompertz equation (*Zwietering et al., 1990*):

$$\log N(t) = \log N_0 + \log \frac{N_{max}}{N_0} \times \exp \left\{ -\exp \left[ \frac{\mu_{max} \times 2.718}{\log(N_{max}/N_0)} \times (Lag - t) + 1 \right] \right\},$$

where $t$ is the time of sampling; $N(t)$ is the cell number of *L. plantarum* 70810 at $t$; $N_0$ and $N_{max}$ are the initial and maximum cell numbers of *L. plantarum* 70810 during soymilk fermentation, respectively; $Lag$ is the lag phase of growth of *L. plantarum* 70810; and $\mu_{max}$ is the specific growth rate of *L. plantarum* 70810. The growth kinetics of *L. plantarum* 70810 was analyzed with Origin software (version9.1; OriginLab, Northampton, MA, USA).

## Determination of pH, titrable acidity (TA), and viscosity of fermented soymilk

To determine pH and TA, 10 g of samples were homogenized with 90 mL of distilled water. pH values were measured by a pH meter (PHS-3C; Shanghai INESA Scientific Instrument Co., Ltd, China.). TA was determined in accordance with AOAC methods (*Chen et al., 2014*). The viscosity values of the fermented and unfermented soymilk were measured directly by a viscometer (NDJ-8S; Shanghai precision electronic instrument Co., Ltd., Shanghai, China).

## Determination of lactic and acetic acids

The concentration of lactic and acetic acids in fermented soymilk were determined as previously described with some modifications (*Teh et al., 2010*). Soymilk was sampled at time 0, 4 and 8 h, after centrifuged at 12,000× g for 15 min, the supernatant was filtered through a filter paper (Whatman 1; Whatman, Maidstone, UK). To precipitate residual proteins, 1 mL of supernatant was added with 0.1 mL of nitric acid (15.8 N) and 0.1 mL of sulphuric acid (0.1 N). The aliquots were filtered through a 0.20 μm filter (Sartorius, Goettingen, Germany). Acetic and lactic acid contents were determined by HPLC, equipped with a Rezex ROA-organic acid H 300 × 7.80 mm column

(Phenomenex, Torrance, CA, USA), a HPLC Pump (Waters, Milford, Ireland) and a 2487 Dual k Absorbance Detector (Waters, Milford, Ireland). Operational conditions were as follows: mobile phase, Sulphuric acid (0.001 N) (Sigma); flow rate, 1.0 mL/min; column temperature, 40 °C; pressure, 500 psi; detector wavelength, 254 nm. Reference acetic and lactic acids (Sigma) were chromatographed to determine their retention times, integrator response factor and recovery value.

## Isoflavone extraction and HPLC analysis

Isoflavone extraction from fermented and unfermented soymilk was performed as previously described with some modifications (*Wei et al., 2007*). Soymilk (10 mL) was dried with a vacuum freeze dryer (SJIA-10N; Ningbo YinZhou Sjia Lab Equipment Co., Ltd., China). Dried samples were mixed with 80 mL of 80% methanol, stirred at 60 °C for 1 h, and then filtered with a Whatman No. 1 filter. The filtrate was dried in a rotary evaporator, redissolved in 50% methanol, and then extracted with 20 mL of n-hexane. All samples were condensed to approximately 1 mL, redissolved in 80% methanol to a final volume of 10 mL, and then passed through a 0.45 µm filter for HPLC analysis. The stock solutions of each of the standard compounds of daidzin, genistin, daidzein and genistein (sigma) were prepared by dissolving 1 mg of each in 10 ml of 80% aqueous methanol and were stored in the refrigerator.

The HPLC system is composed of a detector (UV-2070; Jascoint, Tokyo, Japan), a pump (PU-2089; Jascoint, Tokyo, Japan), and a C18 packed column (Vydac 218TP54, 4.6 mm × 250 mm, 5 µm Spherical; Grace Vydac, Hesperia, CA, USA). Solvent A was acetonitrile, and solvent B was water containing 1% trifluoroacetic acid (v/v). The flow rate was set to 0.8 mL /min, column temperature was set to 20 °C, and an UV–Vis detector was set at 260 nm. A gradient solvent system was applied after injecting 20 µL of the sample into the HPLC system. At 0–6 min, solvent A increased from 10% to 20% while solvent B decreased from 90% to 80%. At 6–30 min, solvent A increased from 20% to 40% while solvent B decreased from 80% to 60%. At 30–35 min, solvent A decreased from 40% to 10% while solvent B increased from 60% to 90%.

## *In vitro* simulated gastric and intestinal stresses tolerance tests

Acidic conditions were simulated by acidic MRS broth with pH adjusted to 3.5, 2.5, and 1.5 by adding 1 M HCl (*Minelli et al., 2004*). Simulated gastric juices were prepared fresh daily by suspending pepsin (Sigma-Aldrich, Poole, UK) (3 g/L) in sterile saline and adjusting pH to 1.5 with 1 M HCl at 37 °C (*Charteris et al., 1998*). Simulated pancreatic juices were prepared fresh daily by suspending pancreatin USP (Sigma-Aldrich) (1 g/L) in sterile saline (0.5% NaCl w/v) with pH adjusted to 8.0 by adding 0.1 M NaOH at 37 °C (*Charteris et al., 1998*). Simulated bile salt solution was prepared by adding 0.1%, 0.2%, or 0.3% (w/v) bile salt (Sigma-Aldrich) to MRS broth.

To test simulated gastric and intestinal stresses resistance, 1 mL of fermented soymilk containing FL or IL (cell counts adjusted to approximately 9 log CFU/g) was incubated in the prepared acidic MRS broth, simulated gastric juices, pancreatic juices, and bile salt solution for 1 or 3 h at 37 °C. Survival was evaluated by plate count on MRS agar.

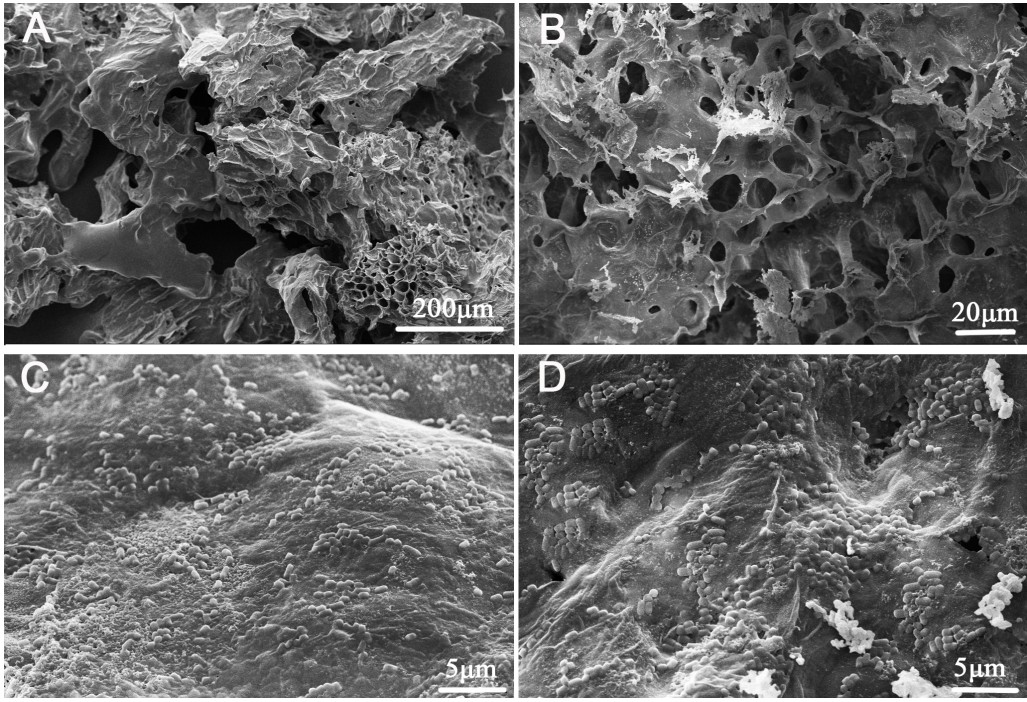

**Figure 1** **Scanning electron micrographs of okara and okara-immobilized *L. plantarum* 70810 cells.** (A, B), portrait slice of okara; (C, D), okara-immobilized *L. plantarum* 70810.

## Statistical analysis

All treatments were performed in triplicate, and data were expressed as means $\pm$ SD. Data were analyzed with general linear model procedures and Duncan's new multiple range tests for comparison of means by SPSS Inc. software (version 15.0) (Chicago, IL, USA) for Windows. A probability of less than 5% ($p \leq 0.05$) was considered statistically significant.

## RESULTS AND DISCUSSION

### Scanning electron microscopy (SEM)

As shown in Fig. 1, a large number of *L. plantarum* 70810 cells attached to the okara's matrices (Figs. 1C and 1D). SEM micrographs showed that the cells remained adhered onto the okara's surface despite excessive washings, indicating successful immobilization. We also found that ultrasonic technology had to be applied to shed off the cells from okara (Fig. S1). We postulate that cell immobilization occurred by covalent binding or physical adsorption by electrostatic forces between *L. plantarum* 70810 cells and okara or by cell entrapment into the vacuous and porous structures found on okara (Figs. 1A and 1B). These structures could provide additional areas for cell adhesion and facilitate mass transportation (*Yu et al., 2007*). Previous studies documented that processes such as grinding, boiling, and sterilization can produce uneven structures that increase available surface areas for cell adsorption (*Raghavendra et al., 2006*; *Bosnea et al., 2009*). Such structures allow bacteria to attach more easily and firmly to immobilizers compared with smooth structures. Other studies on immobilization have reported this
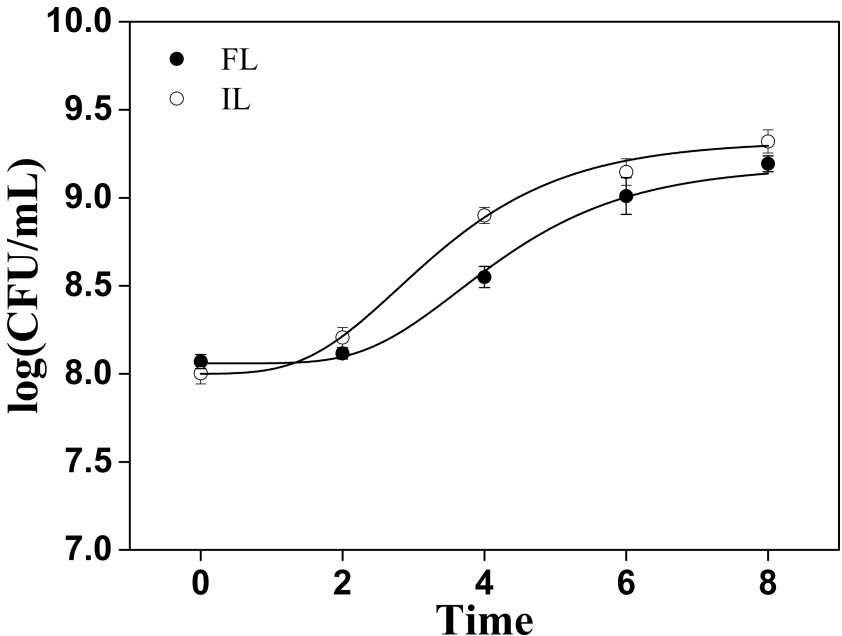

**Figure 2** Cells count changes in soymilk containing free and okara-immobilized *L. plantarum* 70810. FL, free *L. plantarum* 70810; IL, okara-immobilized *L. plantarum* 70810; CFU: colony forming units.

phenomenon (*Yu et al., 2010*; *Genisheva et al., 2011*). *Espinosamartos & Rupérez (2009)* reported that undigestible carbohydrates (mainly insoluble dietary fibre) and protein, the main components of okara, could be fermented by beneficial species, and this report promoted us to postulated that *L. plantarum* 70810 may preferentially bind to specific carbohydrate/protein sites.

### Growth conditions of *L. plantarum* 70810 in soymilk

Changes in microbial counts of soymilk inoculated with FL or IL are shown in Fig. 2. The initial count numbers for FL or IL in soymilk were not significantly different at $8.8 \pm 0.5$ and $8.02 \pm 0.06$ log CFU/mL, respectively. FL and IL in soymilk proliferated after a lag phase of growth. However, the growth of IL in soymilk was faster compared with that of FL in soymilk. Further analysis showed that the specific growth rate of IL in soymilk was $0.37 \pm 0.02 \text{ h}^{-1}$, whereas that of FL in soymilk was $0.31 \pm 0.03 \text{ h}^{-1}$. Meanwhile, the lag phase of growth of IL in soymilk lasted for $1.47 \pm 0.13$ h, whereas that of FL in soymilk lasted for $2.30 \pm 0.23$ h . Lemons, oranges, agro-wastes, and cereals contain high amounts of dietary fibres, sugars, minerals, and essential vitamins that facilitate the growth of probiotics (*Charalampopoulos et al., 2003*; *Sendra et al., 2008*; *Teh et al., 2010*). It also has been reported that okara comprised 14.5–55.4% dietary fiber, 24.5–37.5% proteins, 9.3–22.3% lipids, and amounts of sugars, minerals and essential vitamins (*O'Toole, 1999*; *Jiménez-Escrig et al., 2008*; *Redondo-Cuenca Villanueva-Suárez & Mateos-Aparicio, 2008*). These ingredients give the okara of potential prebiotic effect which benefit the growth of probiotics (*Pérezlópez et al., 2016*). We postulated that IL exhibits a faster growth rate and

**Table 1** The changes in lactic and acetic acids of soymilk inoculated with free and immobilized *L. plantarum* 70810.

| Time(h) | Lactic acid contents (mg/ml) | | Acetic acid contents (mg/ml) | |
|---|---|---|---|---|
| | FL | IF | FL | IL |
| 0 | $0.93 \pm 0.04$ | $0.95 \pm 0.04$ | $0.98 \pm 0.07$ | $1.02 \pm 0.07$ |
| 4 | $1.77 \pm 0.07^{**}$ | $2.05 \pm 0.07^{*,**}$ | $1.24 \pm 0.06^{**}$ | $1.44 \pm 0.77^{*,**}$ |
| 8 | $4.69 \pm 0.18^{**}$ | $5.32 \pm 0.10^{*,**}$ | $1.90 \pm 0.10^{**}$ | $2.22 \pm 0.10^{*,**}$ |

**Notes.**

FL, free *L. plantarum* 70810; IL, okara-immobilized *L. plantarum* 70810.

[*] $p < 0.05$ vs. free *L. plantarum* 70810;

[**] $p < 0.05$ vs. time 0.

shorter lag phase because of the availability of fibers, proteins, lipids, sugars, minerals, and essential vitamins in okara.

The decrease in pH and increase in TA and viscosity were accompanied by *L. plantarum* 70810 growth. Table 1 and Fig. 3 show the differences in the concentrations of lactic and acetic acids, pH, TA, and viscosity between soymilk inoculated with FL and that inoculated with IL. During fermentation, a lower pH and higher concentrations of lactic and acetic acids, TA and viscosity were observed in soymilk inoculated with IL compared with that inoculated with FL because of the higher growth rate and shorter lag phase of growth of IL than FL. The concentration of lactic acid was found higher than acetic acid in both cultures fermented by IL and FL over 8 h, and the concentrations of lactic and acetic acids were higher in soymilk fermented by IL than FL (Table 1). IL culture acidified soymilk to pH 4.5 (end-point of fermentation) in approximately 6 h, whereas FL culture acidified soymilk to pH 4.5 in approximately 8 h (Fig. 3A). Consistently, soymilk inoculated with IL attained a TA of 50 °T in 6 h, whereas soymilk inoculated with FL attained a TA of 50 °T in 8 h (Fig. 3B). These results agreed with those reported by *Kourkoutas et al. (2005)* and *Kourkoutas et al. (2006)*, who found that immobilized probiotic bacteria on fruit segments (apple and pear) showed a faster rate of pH decrease and a lower final pH upon reactivation in whey. Our results also agreed with those reported by *Teh et al. (2010)*, who found that immobilized lactobacilli show significantly better growth ($P < 0.05$) compared with free lactobacilli and that growth is accompanied by a higher production of lactic and acetic acids in soymilk, resulting in a lower final pH.

The viscosity of soymilk inoculated with IL increased significantly faster than that of soymilk inoculated with FL (Fig. 3C). This finding might be mainly due to IL's higher growth rate and higher substrate utilization than FL, leading to the increased production of organic acids (mainly lactic and acetic acids), which decreased the pH of soymilk, and precipitation was induced when the pIs of the soy proteins was reached (*Pyo et al., 2005*; *Liu et al., 2009*; *Grygorczyk & Corredig, 2013*; *Chen et al., 2014*). Moreover, the polysaccharides and their hydrolyzates in okara might also play an important role in the increase of viscosity of soymilk inoculated with IL (*Villanueva-Suárez et al., 2013*).

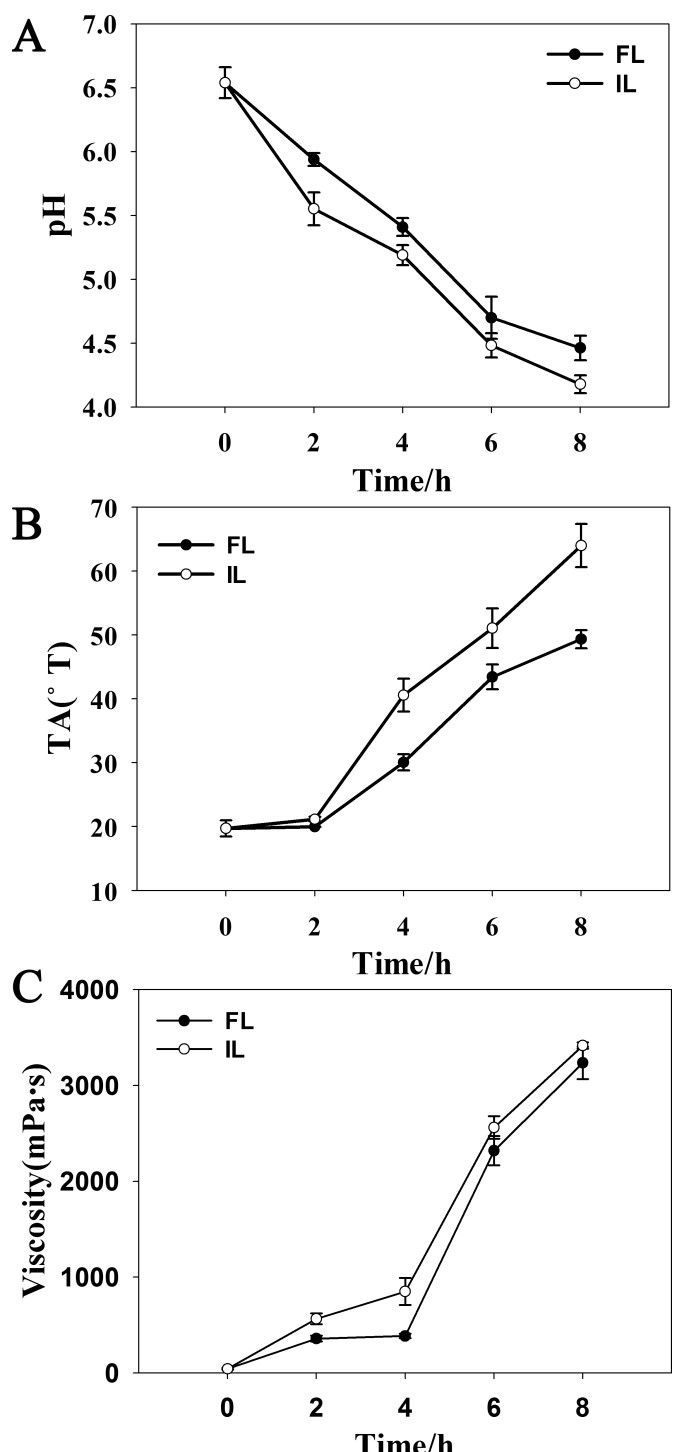

**Figure 3** **Fermentation parameters during soymilk fermentation using free and okara-immobilized *L. plantarum* 70810.** FL, free *L. plantarum* 70810; IL, okara-immobilized *L. plantarum* 70810; TA, Titrable Acidity; ° T, Thorner degree.
Table 2  The change in soybean isoflavone content of soymilk inoculated with free and immobilized *L. plantarum* 70810.

| Time(h) | Glucosides | | | | Aglycones | | | |
|---|---|---|---|---|---|---|---|---|
| | Daidzin (µg/mL) | | Genistin (µg/mL) | | Daidzein (µg/mL) | | Genistein (µg/mL) | |
| | FL | IL | FL | IL | FL | IL | FL | IL |
| 0 | 44.56 ± 4.66 (37.91%) | 45.31 ± 5.32 (37.80%) | 65.61 ± 5.17 (55.82%) | 67.15 ± 5.09 (56.02%) | 3.07 ± 0.21 (2.62%) | 3.08 ± 0.27 (2.57%) | 4.29 ± 0.38 (3.68%) | 4.33 ± 0.31 (3.61%) |
| 4 | 29.57 ± 1.68[**] (29.05%) | 20.87 ± 1.98[**] (22.05%) | 44.55 ± 3.16[**] (43.76%) | 28.93 ± 2.01[**,*] (30.57%) | 11.81 ± 0.85[**] (11.60%) | 17.25 ± 1.34[**,*] (18.23%) | 15.87 ± 1.16[**] (15.59%) | 27.59 ± 2.07[**,*] (29.15%) |
| 8 | 12.83 ± 0.84[**] (13.92%) | 5.85 ± 0.57[**,*] (6.62%) | 19.93 ± 0.78[**] (21.63%) | 6.91 ± 0.53[**,*] (7.82%) | 24.62 ± 1.76[**] (26.72%) | 31.19 ± 2.21[**,*] (35.29%) | 34.75 ± 2.21[**] (37.72%) | 44.42 ± 3.16[**,*] (50.27%) |

Notes.

FL, free *L. plantarum* 70810;  IL, okara-immobilized *L. plantarum* 70810.

[*]$p < 0.05$ vs. free *L. plantarum* 70810.

[**]$p < 0.05$ vs. time 0.

## Isoflavone compositions in fermented soymilk

Soybean is rich in isoflavones, which exhibit weak estrogen activity, act as antioxidants, prevent osteoporosis and cancer, reduce total cholesterol, delay menopause, and provide other health benefits (*Chang & Nair, 1995*). Aglycones and the glucosidic conjugates are the basic categories of isoflavones. In unfermented soybean products, daidzin and genistin are the main glucosidic isoflavones, which comprise 80%–95% of total isoflavones, and daidzein and genistein are the main aglycones (*Coward et al., 1998*). Many studies have indicated that the biological effects of isoflavones are conferred by aglycones and not glycosides (*Kawakami et al., 2005*); thus, isoflavone glucosides must be hydrolyzed to have a biological effect. $\beta$-glucosidase can hydrolyze glucoside isoflavones with the formation of aglycones (*Esaki et al., 2004*). Probiotics with $\beta$-glucosidase can increase aglycone content during soymilk fermentation (*Martinezvillaluenga et al., 2012*).

To compare the effect of FL and IL on the bioconversion of daidzin and genistin, they were inoculated in soymilk and the four isoflavones, daidzin, genistin, daidzein, and genistein were analyzed. As shown in Table 2, in unfermented soymilk, the contents of daidzin and genistin were up to 37.91% and 55.82%, while those of daidzein and genistein were only 2.62% and 3.68%. Soymilk fermented with FL and IL exhibited a drastic reduction in daidzin and genistin contents and a drastic increase in daidzein and genistein contents. The daidzin and genistin contents in soymilk fermented with FL decreased to 29.05% and 43.76% after 4 h of incubation and to 13.92% and 21.63% after 8 h of incubation. The daidzin and genistin contents decreased to 22.05% and 30.57% after 4 h of incubation and to 6.62% and 7.82% after 8 h of incubation in soymilk fermented with IL. The daidzein and genistein contents in soymilk fermented with FL increased to 11.60% and 15.59% after 4 h of incubation and to 26.72% and 37.72% after 8 h of incubation, whereas those in soymilk fermented with IL increased to 18.23% and 29.15% after 4 h of incubation and to 35.29% and 50.27% after 8 h of incubation. These results indicated that fermentation of soymilk by IL caused a faster reduction in daidzin and genistin contents and a faster increase in their respective aglycones compared with FL. This difference might be attributed to the faster growth rate of IL in soymilk compared with FL.

**Table 3** Effect of acidic conditions on the survival of free and immobilized *L. plantarum* 70810 (log (CFU/g)).

| Time (h) | pH 3.5 | | pH 2.5 | | pH 1.5 | |
|---|---|---|---|---|---|---|
| | FL | IL | FL | IL | FL | IL |
| 0 | $9.06 \pm 0.18$ | $9.14 \pm 0.18$ | $9.06 \pm 0.18$ | $9.14 \pm 0.18$ | $9.06 \pm 0.18$ | $9.14 \pm 0.18$ |
| 1 | $8.26 \pm 0.16^{**}$ | $8.43 \pm 0.17^{**}$ | $8.01 \pm 0.15^{**}$ | $8.25 \pm 0.14^{**}$ | $6.73 \pm 0.13^{**}$ | $7.37 \pm 0.15^{**,*}$ |
| 3 | $8.19 \pm 0.21^{**}$ | $8.42 \pm 0.16^{**}$ | $7.62 \pm 0.16^{**}$ | $8.19 \pm 0.21^{**,*}$ | $4.88 \pm 0.10^{**}$ | $6.21 \pm 0.10^{**,*}$ |

**Notes.**
FL, free *L. plantarum* 70810; IL, okara-immobilized *L. plantarum* 70810.
$^{*}p < 0.05$ vs. free *L. plantarum* 70810.
$^{**}p < 0.05$ vs. time 0.

**Table 4** Effect of simulated gastric transit and pancreatic juice on the survival of free and immobilized *L. plantarum* 70810 (log (CFU/g)).

| Time (h) | Simulated gastric juice | | Simulated pancreatic juice | |
|---|---|---|---|---|
| | FL | IL | FL | IL |
| 0 | $9.05 \pm 0.19$ | $9.16 \pm 0.21$ | $9.05 \pm 0.19$ | $9.16 \pm 0.21$ |
| 1 | $6.61 \pm 0.15^{**}$ | $7.17 \pm 0.13^{**,*}$ | $7.31 \pm 0.09^{**}$ | $8.29 \pm 0.11^{**,*}$ |
| 3 | $4.75 \pm 0.13^{**}$ | $6.01 \pm 0.15^{**,*}$ | $7.19 \pm 0.13^{**}$ | $8.19 \pm 0.15^{**,*}$ |

**Notes.**
FL, free *L. plantarum* 70810; IL, okara-immobilized *L. plantarum* 70810.
$^{*}p < 0.05$ vs. free *L. plantarum* 70810.
$^{**}p < 0.05$ vs. time 0.

**Table 5** Effect of bile salts on the survival of free and immobilized *L. plantarum* 70810 (log (CFU/g)).

| Time (h) | 0.1% | | 0.2% | | 0.3% | |
|---|---|---|---|---|---|---|
| | FL | IL | FL | IL | FL | IL |
| 0 | $9.05 \pm 0.19$ | $9.16 \pm 0.21$ | $9.05 \pm 0.19$ | $9.16 \pm 0.21$ | $9.05 \pm 0.19$ | $9.16 \pm 0.21$ |
| 1 | $9.00 \pm 0.18$ | $9.10 \pm 0.16$ | $5.31 \pm 0.11^{**}$ | $7.99 \pm 0.14^{**,*}$ | $4.18 \pm 0.09^{**}$ | $7.38 \pm 0.11^{**,*}$ |
| 3 | $8.87 \pm 0.14$ | $9.06 \pm 0.17$ | $4.71 \pm 0.10^{**}$ | $7.19 \pm 0.15^{**,*}$ | $3.94 \pm 0.07^{**}$ | $6.29 \pm 0.13^{**,*}$ |

**Notes.**
FL, free *L. plantarum* 70810; IL, okara-immobilized *L. plantarum* 70810.
$^{*}p < 0.05$ vs. free *L. plantarum* 70810.
$^{**}p < 0.05$ vs. time 0.

## Simulated gastric and intestinal stresses tolerance tests

To estimate cell tolerance to the simulated gastric and intestinal conditions, fermented soymilk containing either FL or IL was exposed to *in vitro* conditions simulating acidic environment, gastric and pancreatic juices, and bile salts; the results are summarized in Tables 3–5. The initial number of colonies used for these tests was estimated at $10^9$ CFU/mL. The log CFU/mL values of soymilk containing FL or IL used to test for low pH tolerance were $9.06 \pm 0.18$ and $9.14 \pm 0.18$, respectively. The log CFU/mL values of soymilk containing FL or IL used to test for tolerance to simulated gastric juice, pancreatic juice, and bile salts were $9.05 \pm 0.19$ and $9.16 \pm 0.21$, respectively. These values did not differ significantly.

### Simulated gastric acid tolerance

Acid tolerance for probiotics is essential not only for resistance to gastric stress, but it is also a prerequisite in the production of acidic probiotic food products. The buffering capacity of the food, which is a major factor affecting pH, and the rate of gastric emptying may significantly influence cell survival in the GI tract (*Kourkoutas et al., 2005*; *Kourkoutas et al., 2006*). Gastric juice pH is one of the main factors determining the survival of probiotic bacteria when passing through the stomach to the intestine. As shown in Table 3, the number of viable cells significantly reduced after 1 and 3 h at pH 3.5 and after 1 h at pH 2.5, but no significant difference in viable cell count was observed between soymilk containing FL and IL. When the incubation time was prolonged to 3 h at pH 2.5, cells in soymilk containing IL showed a significantly higher survival level compared with the cells in soymilk containing FL. The number of viable cells drastically reduced in soymilk containing FL or IL when the pH of the MRS broth decreased to 1.5 to simulate the extreme pH conditions of the stomach. However, IL in soymilk exhibited a significantly higher viability compared with FL in soymilk. The number of viable cells in soymilk containing FL decreased by 2.43 and 4.26 log cycles after 1 and 3 h, respectively, whereas the number of viable cells in soymilk containing IL decreased by 1.77 and 2.93 log cycles after 1 and 3 h, respectively. IL in soymilk also had a significantly higher final viable count than FL in soymilk. Tolerance to upper GI transit was also predicted with simulated gastric juice (pH 1.5, Table 4). Table 3 shows that the viable cell number of FL in soymilk was reduced by 2.44 and 4.30 log cycles after incubation for 1 h and 3 h in simulated gastric juice, whereas the number of viable cells in soymilk containing IL at the both time points was reduced by only 1.99 and 3.15 log cycles. IL in soymilk also showed significantly higher cell survival rates. These results coincided with those obtained from MRS broth with pH 1.5, indicating that pH is the main factor affecting probiotics survival in the stomach. Our results also agreed with other studies. *Sidira et al. (2010)* reported that acidic conditions significantly reduce the number of both free and immobilized *L. casei* ATCC 393 cells. However, the count number of immobilized cells is significantly higher than free cells after 120 min at pH 2.0 and after 30, 60, 90, and 120 min at pH 1.5. *Mokarram et al. (2009)* showed that cell viability is reduced by three log cycles when calcium alginate capsules containing *L. acidophilus* are incubated in simulated gastric juice (pH 1.5), whereas coating the capsules with one or two layers of sodium alginate improves cell survival by one and two log cycles, respectively. *Laelorspoen et al. (2014)* incubated cells encapsulated in alginate and citric acid-modified zein coating in gastric fluid (pH 1.2) at 37 °C for 2 h and obtained cell counts of 7.14 log CFU/mL compared with 4.52 log CFU/mL for free-cell suspensions. *Fijałkowski & Peitler (2015)* found that the viability of *Lactobacillus* cells adsorbed on or entrapped in bacterial cellulose incubated in simulated gastric juices for 4 h is significantly higher than that of free cells, particularly for *Lactobacillus* cells entrapped in bacterial cellulose showed a viability more than 70% compared with less than 10% for free cells, and our research can well support their finding, becuse the insoluble dietary fibre, the main composition of okara, is mainly composed by cellulose (*Redondo-Cuenca Villanueva-Suárez & Mateos-Aparicio, 2008*; *Espinosamartos & Rupérez, 2009*).

*Simulated pancreatic juice tolerance*

We also studied the survival of FL and IL in soymilk in simulated pancreatic juice (Table 4). Our results showed that simulated pancreatic juice significantly reduced the survival of both FL and IL. The number of viable cells in soymilk containing IL was significantly higher at both time points compared with that in soymilk containing FL. This result differed from those reported by *Sidira et al. (2010)*, who found that simulated pancreatic juice exerts no effect on the survival of immobilized *L. case* ATCC399 but significantly affects the viability of free *L. case* ATCC399. This result might be attributed to the use of different strains or support materials.

*Simulated bile salt tolerance*

Our work also assessed simulated bile salt tolerance. As shown in Table.5, 0.1% bile salt exerted no significant effect on the survival of FL or IL in soymilk. When the concentration of bile salt increased to 0.2% and 0.3%, the survival of FL or IL in soymilk significantly decreased. However, IL showed a significantly higher number of viable cells compared with FL. These results were in line with the observations of *Sidira et al. (2010)*. In their study, the viable cell count of *L. case* ATCC399 immobilized in apple pieces decreased from 9.30 log CFU/mL to 6.23 log CFU/mL after 4 h of incubation in 1% bile salt solution, whereas the viable cell count of free *L. case* ATCC399 decreased from 9.16 log CFU/mL to 3.66 log CFU/mL. *Michida et al. (2006)* fond that *L. plantarum* NCIMB 8826 immobilized within malt and barley cereal fiber could improve its viability in bile salt solution. According to these results, we postulated that IL showed improved tolerance to bile salt stress compared with FL might be due to the physical entrapment of bile salts into the dietary fibre of okara.

## CONCLUSIONS

(1) Soybean residue (okara) is a food-grade-quality, cheap, and abundant cell support. Okara is as a perfect support material for cell immobilization. Cells are firmly and easily immobilized onto okara because of its vacuous and porous structure.

(2) IL cells showed a faster growth rate and a shorter lag phase of growth in soymilk.

(3) Soymilk inoculated with IL showed higher productions of lactic and acetic acids, a faster decrease in pH and increase in acidity and viscosity compared with soymilk inoculated with FL.

(4) IL accelerated the bioconversion of glucosidic isoflavones to aglycone isoflavones compared with FL during soymilk fermentation.

(5) IL exhibited a significantly enhanced resistance to the simulated gastric and intestinal stresses compared with FL.

(6) Okara used as an immobilizer not only could increase the production rate of probiotics and benefit human health but could also alleviate environmental and economic issues by reducing waste accumulation. Utilizing okara as an immobilization support will also benefit the agricultural industries by providing a sustainable approach in waste management.

### Funding
This study was financially supported by the National Natural Science Foundation of China (Grant No. 31501460). The funders had no role in study design, data collection and analysis, decision to publish, or preparation of the manuscript.

### Grant Disclosures
The following grant information was disclosed by the authors:
National Natural Science Foundation of China: 31501460.

### Competing Interests
The authors declare there are no competing interests.

### Author Contributions
- Xia Xiudong conceived and designed the experiments, performed the experiments, analyzed the data, wrote the paper, prepared figures and/or tables.
- Wang Ying contributed reagents/materials/analysis tools.
- Liu Xiaoli and Li Ying reviewed drafts of the paper.
- Zhou Jianzhong conceived and designed the experiments.

### Data Availability
   The raw data has been supplied as Supplemental File.

### Supplemental Information
Supplemental information for this article can be found online at http://dx.doi.org/10.7717/peerj.2701#supplemental-information.

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
