# Peer review of "Soymilk residue (okara) as a natural immobilization carrier for Lactobacillus plantarum cells enhances soymilk fermentation, glucosidic isoflavone bioconversion, and cell survival under simulated gastric and intestinal conditions"

_PeerJ, doi:10.7717/peerj.2701_

## Round 0.1 · original submission · Major Revisions

Please consider all the questions raised up by the reviewers (and note that Reviewer 2 has also provided detailed comments in an attached PDF). You do not have to include all the modifications they suggested, but if you decide to discard some of them, an explanation should be provided in the rebuttal letter.

Reviewer 1 ·

Basic reporting

No comments

Experimental design

The experimental part was designed properly.

Validity of the findings

No comments

Additional comments

The paper investigates immobilization of Lactobacillus plantarum onto soymilk residue (Okara) and examines the growth and metabolic characteristics of immobilized cells in soymilk. In addition, the survival of immobilized L. plantarum cells under simulated gastrointestinal conditions is assessed.
Although of limited novelty, it is an interesting, well-written and important for the field study an thus I suggest its publication.

Please mind the following minor comments:

l. 230. “of soymilk” is written twice.
l. 247-249. Incorrect English. Please revise. Generally, English language should be significantly improved. Please seek advice from an English native speaker.

Reviewer 2 ·

Basic reporting

In my opinion, the manuscript would greatly improve if more emphasis was put on the prebiotic role played by okara, a unique substrate for the immobilization of probiotic lactobacilli, such as L. plantarum.
To revise the manuscript please take into account the specific comments given as an attachment.

Experimental design

It is appropriate and results are clearly presented and easy to follow. Regarding the most relevant results found: IL cells promoted a higher specific growth rate on okara substrate, as well as higher acidity and viscosity of fermented soymilk within a shorter time. Similarly, IL gave a higher aglycones content in fermented soymilk, as compared to the control. Further, IL showed better resistance to simulated gastrointestinal (GI) conditions at acidic pH, or when pepsin, pancreatin and bile salts were added.

Validity of the findings

Okara is proposed as a potential new carrier for L. plantarum immobilization, capable to improve cell growth and bioconversion of glucosides into the corresponding aglycone isoflavones in fermented soymilk, as well as to preserve cell survival and viability through the gut.

Additional comments

This is an interesting article on the potential use of soybean okara as a natural, efficient and cheap immobilizer for probiotic lactic acid bacteria (L. plantarum).
Okara -the main food-grade by-product from soymilk production- used in this work has been prepared by the Chinese procedure, in which no thermal treatment is applied to the blended mixture of soybean seeds. It is highly desirable to improve the survival and viability of probiotics along the gut and, for this purpose, cell immobilization is one of the most promising techniques. Moreover, bioconversion of glucosidic isoflavones (daizin and genistin) into aglycones (daizein and genistein), and cell resistance to gastrointestinal stress conditions during soymilk fermentation were studied in vitro on okara-immobilized L. plantarum (IL), in comparison to the free cells (FL) control.

Annotated reviews are not available for download in order to protect the identity of reviewers who chose to remain anonymous.

Reviewer 3 ·

Basic reporting

The article is written in correct English, with a good structure. Table 1 should be removed and data can be presented only in the text. Figures are relevant, with good resolution and properly labeled, although Figure 1 scale has a poor visibility. Also please, check the scale of figure 1A, 1B, 1C and 1D, in my experience it could be an error among the scales.

Experimental design

On my own knowledge, this is an original contribution, the hypothesis is clear defined and research questions are quite well stated. The experiments were conducted to fully answered the aim of the work and fulfill the appropriate technical standards. Nevertheless, there are some issues that should be improved:
1. Line 174, 202: Units used for growth: among the text, why did you use log (CFU/g) for growth when most of the experiments were run in liquid medium. Further all the methods described in the work are referred to volumes of soy milk, medium, etc.. but them the results for growth is presented “per gram” … gram of what?? I think that authors should recalculate growth according to the experiments media, not only for further understanding of the results but also for allowing the comparison with the data they are showing during the discussion of the different items (all the referred data are in log (CFU/mL) ( line 303, 320-322).
2. Please check that the same nomenclature is used elsewhere in the text. i.e: line 267, in the same line authors used to different letters for milliliters: ml or mL.

3. Authors stated that they had “assessed the survival of okara-immobilized L.plantarum cells under simulated gastrointestinal conditions”(line 81), although what they really do is asses three intestinal conditions separately, why do not assay the survival in a coupled experiment of the three conditions?? I think that is statement is confused and should be misunderstand, so although in the past, some authors had stated these kinds of separate experiments as GI conditions. These should not be considered as a GI tract conditions simulation. I think that this should be rephrase to clarify among the text (lines 18, 24, 27, 81, 165-176, 263-324, 334) or it should be included as a limitation in the conclusion section.

Validity of the findings

Data are robust and controlled, although there is no mention about the statistical analysis performed for the Figure 2 and 3, either in the methods or in the results section.
Additionally the statement about the simulations of GI tract conditions is inadequate, as mentioned in the above section: they stated that had “assessed the survival of okara-immobilized L.plantarum cells under simulated gastrointestinal conditions”(line 81). Thus, it should be rephrase to clarify among the text or included as a limitation in the conclusion section.

Additional comments

The article is written in correct English, with a good structure. This is an original contribution, the hypothesis is clear defined and research questions are quite answered. The experiments were conducted answered the aim of the work and fulfill the appropriate technical standards unless for that of the survival simulation throughout the GI tract.

---

## Round 0.2 · accepted · Accept

I think the authors have properly considered all the comments raised up by the reviewers, making the proper modifications or additions in the manuscript.